# Architecture of a SARS-CoV-2 mini replication and transcription complex

Liming Yan[1,5], Ying Zhang[2,5], Ji Ge[2], Litao Zheng[2], Yan Gao[1,3], Tao Wang[1], Zhihui Jia[4], Haofeng Wang[3], Yucen Huang[1], Mingyu Li[1], Quan Wang [3], Zihe Rao[1,2,3 ✉] & Zhiyong Lou [1 ✉]

Non-structural proteins (nsp) constitute the SARS-CoV-2 replication and transcription complex (RTC) to play a pivotal role in the virus life cycle. Here we determine the atomic structure of a SARS-CoV-2 mini RTC, assembled by viral RNA-dependent RNA polymerase (RdRp, nsp12) with a template-primer RNA, nsp7 and nsp8, and two helicase molecules (nsp13-1 and nsp13-2), by cryo-electron microscopy. Two groups of mini RTCs with different conformations of nsp13-1 are identified. In both of them, nsp13-1 stabilizes overall architecture of the mini RTC by contacting with nsp13-2, which anchors the 5′-extension of RNA template, as well as interacting with nsp7-nsp8-nsp12-RNA. Orientation shifts of nsp13-1 results in its variable interactions with other components in two forms of mini RTC. The mutations on nsp13-1:nsp12 and nsp13-1:nsp13-2 interfaces prohibit the enhancement of helicase activity achieved by mini RTCs. These results provide an insight into how helicase couples with polymerase to facilitate its function in virus replication and transcription.

[1] MOE Key Laboratory of Protein Science, School of Medicine, Tsinghua University, Beijing, China. [2] School of Life Sciences, Tsinghua University, Beijing, China. [3] Shanghai Institute for Advanced Immunochemical Studies and School of Life Science and Technology, ShanghaiTech University, Shanghai, China. [4] Research Centre of Microbiome, Department of Medical Microbiology and Parasitology, School of Basic Medical Sciences, Capital Medical University, Beijing, China. [5]These authors contributed equally: Liming Yan, Ying Zhang. ✉email: raozh@tsinghua.edu.cn; louzy@mail.tsinghua.edu.cn

Severe acute respiratory syndrome coronavirus 2 (SARS-CoV-2) is the causative agent of the pandemic of coronavirus disease 2019 (COVID-19)[1–6]. According to the World Health Organization (WHO) on August 6, 2020, 2.04 million infections and over 744 thousand deaths have been confirmed globally.

SARS-CoV-2 encodes 16 non-structural proteins (nsp) to assemble a so-called replication and transcription complex (RTC) to facilitate virus replication and transcription[7]. Among them, nsp12 functions as RNA-dependent RNA polymerase (RdRp) by using nsp7 and nsp8 as its auxiliary factors, and nsp13 is identified as a helicase. Nsp13 catalyzes the unwinding of duplex oligonucleotides into single strands in an ATP-dependent manner, and its helicase catalytic rate can be enhanced by the presence of polymerase[8,9]. The structural studies of nsp13 encoded by SARS-CoV and MERS-CoV have shown that nsp13 belongs to the SF1 helicase superfamily, with an N-terminal zinc-binding domain (ZBD), two helicase core domains RecA1 (1A) and RecA2 (2A), and an inserted domain 1B being connected to ZBD via a "stalk" region[9,10]. Moreover, nsp13 is also suggested to have a role in mRNA capping and play multiple roles in coronavirus life cycle[11,12]. In the previous works, we have determined the structures of nsp7–nsp8–nsp12 and its complex with template-primer RNA, which is the central component of SARS-CoV-2 RTC (hereafter named central RTC)[13,14]. Here we aim to dissect the mechanism for how helicase couples with central RTC.

## Results

**Assembly of a SARS-CoV-2 mini RTC.** We expressed the full-length SARS-CoV-2 nsp12 (residues S1-Q932), nsp8 (residues A1-Q198) and nsp7 (residues S1-Q83) in *E. coli*, and reconstituted a central RTC with a template-primer RNA which has an additional 5′ extension in the template (Fig. 1a, b and Supplementary Fig. 1). The bacterially expressed and purified nsp13 (residues A1-Q601) was subsequently incubated with nsp7-nsp8-nsp12-RNA to form mini RTC. The helicase activity of apo nsp13 is enhanced by the formation of mini RTC (Fig. 1c), approving this mini RTC is functional, which is consistent with the previous result[8,11]. Cryo-EM grids were prepared, and preliminary screening revealed ideal particle density with good dispersity (Supplementary Fig. 2). After collecting and processing 4107 micrograph movies, we identified two major classes of mini RTC and reconstructed their structures to the resolution of 2.98 Å (form 1, 83,144 particles (75.6% of total particles)) and 3.84 Å (form 2, 26,768 particles (24.4% of total particles)) (Fig. 1d, Supplementary Fig. 2, and Supplementary Table 1).

**The architecture of mini RTC.** In two forms of mini RTC, there is one nsp7, two nsp8 (hereafter named nsp8-1 and nsp8-2 as previously named[13] and indicated in Fig. 1b, d), one nsp12, two nsp13 (hereafter named nsp13-1 and nsp13-2 as indicated in Fig. 1d), and one template-primer RNA (Fig. 1, Supplementary Figs. 3 and 4). The template-primer RNA, nsp7, nsp8, and nsp12 assemble the central component of mini RTC with the same architecture as previously reported[13–15]. Two nsp13 molecules bind on the plain region formed by two nsp8 and nsp12. Nsp13-1 interacts with nsp8-1 and the interface domain of nsp12 via its helicase ZBD and 1B domain, respectively (Supplementary Fig. 5 and Supplementary Table 2). Nsp13-2 ZBD contacts with nsp8-2 helical domain and the thumb domain of nsp12, and the nsp13-2 1B domain makes additional contact with nsp8-2 (Supplementary Fig. 5 and Supplementary Table 2). Two nsp13 molecules have

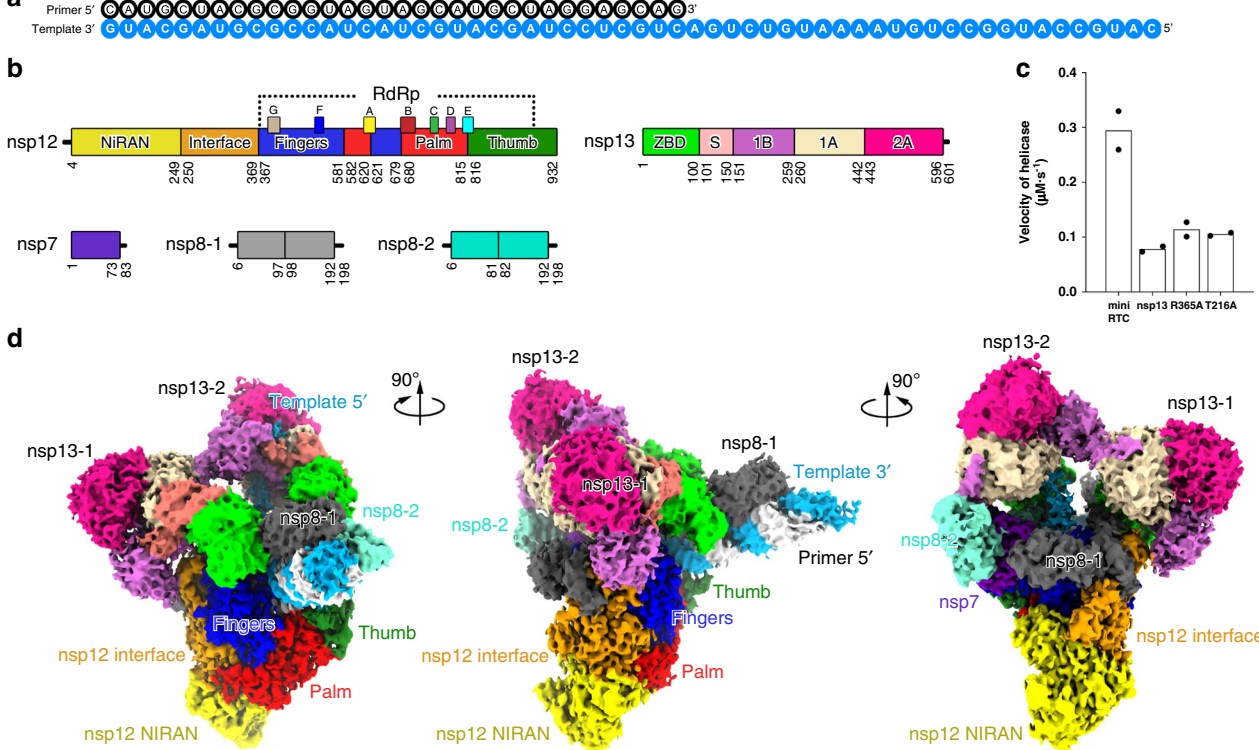

**Fig. 1 Architecture of mini RTC. a** The scaffold of RNA used in the structural study is shown. **b** Domain organization of each component in mini RTC. The color scheme for nsp7, nsp8, and nsp12 are generally the same as those used in our previous work[13] with slight modifications. Domains of nsp13 are shown in different colors with labels. **c** Helicase activities. The helicase activities of mini RTC, the individual nsp13 (nsp13), mini RTC with an R365A mutation in nsp12, and mini RTC with a T216A mutation in nsp13. The velocities of helicase activity are shown as the mean of two independent experiments. **d** Cryo-EM density of mini RTC (represented by form 1) is shown in three perpendicular views. The color scheme is the same as used in (**b**).

interactions with each other mediated by their 1B domains (Supplementary Table 3). The paired portion of template-primer RNA is tightly enwrapped by two nsp8 molecules and nsp12, which is similar to previous structural observation[14,15].

**Interactions of nsp13-2 and template RNA.** The unpaired 5′ extension of RNA template perpendicularly protrudes to nsp13-2 RNA-binding channel from the polymerase catalytic site of nsp12 (Supplementary Fig. 4); whereas, no extra density can be observed around nsp13-1. The cryo-EM densities with good quality allow us to build the nucleosides buried in the RNA-binding channel of nsp13-2, and the RNA region between nsp13-2 and RdRp active site is missing due to insufficient map quality (Fig. 2). For note, because the resolution of form 1 mini RTC is better than that of form 2, we can build six nucleosides of the 5′ extension of RNA template in nsp13-2 in form 1 mini RTC, but only build five nucleosides in nsp13-2 in form 2; nevertheless, the bound 5′ extensions of RNA template in nsp13-2 in two forms are in similar conformation (Supplementary Fig. 4). Hereafter, we use form 1 mini RTC to illustrate the binding of the 5′ extension of RNA template in nsp13-2. In nsp13-2, the bound 5′ extension of RNA template is in an extended conformation and lies in an RNA-binding channel formed by nsp13 1A, 1B, and 2A domains, with a direction of its 5′ end in 2A domain and its 3′ end in 1A domain (Fig. 2a). Key residues involved in nsp13-RNA recognition include N361 in 1A domain, S468/T532/D534 in 2A domain, and R178/H230 in the 1B domain (Fig. 2b).

Consistently, the regions spanning the counterparts of these residues in SARS-CoV nsp13 have been shown to be directly involved in the binding with nucleic acid through H/D exchange mass spectrometry[9] (Supplementary Table 4). Previous structural study on the helicase encoded by equine arteritis virus (EAV, a member of *Arteriviridae* family in *Nidovirales* order) has shown that the 1B domain of the viral helicase has the most significant conformational shift upon substrate binding and play a key role in RNA binding[16] (Supplementary Fig. 6a). Compared to SARS-CoV-2 nsp13 in the apo form (PDB: 6ZSL), 1B domain of nsp13-1 is in a closed conformation, which is similar to apo nsp13; but the 1B domain of nsp13-2 has the shift to allow the entrance of RNA binding channel in a fully open state (Supplementary Fig. 6b).

RecD2 is a member of SF1B with low sequence homology with SARS-CoV-2 nsp13. But the RNA-binding mode of SARS-CoV-2 nsp13 is conserved with that in the SF1B family. Sequence comparison of key residues for RNA binding in SARS-CoV-2/ SARS-CoV/MERS-CoV nsp13 and EAV helicase (nsp9) and RecD2 show that R178 (in 1B domain) and T532 (in 2A domain) are highly conserved. Moreover, H230 (in 1B domain), N361 (in 1A domain), and S486 are conserved in coronaviruses but are different from their counterparts in EAV nsp9 and *Deinococcus radiodurans* RecD2 (Supplementary Fig. 7).

**Structural comparison of two mini RTCs.** Although the overall architectures of two mini RTCs are generally similar, the conformations of nsp13-1 have a distinct shift (Fig. 3a). In the alignment of two forms of mini RTC, nsp7–nsp8–nsp12 and nsp13-2 present a r.m.s.d of 0.429 Å for 1905 Cα atoms, suggesting their highly similar conformations. However, nsp13-1 in form 1 mini RTC has a ~10 Å movement towards nsp13-2, compared with they are in form 2 mini RTC. In mini RTC, nsp13 molecules have three major contacting regions (regions 1, 2, and 3) and two minor regions (regions 4 and 5) with nsp7–nsp8–nsp12 or with each other (Fig. 3b and Supplementary Fig. 8). Region 1 includes the interaction between nsp13-1 1B domain and nsp12 Interface; region 2 includes the interaction

between nsp13-1 ZBD domain and nsp8-1 helical domain; region 3 includes the 1B interaction between nsp13-1 and nsp13-2; region 4 includes one main chain hydrogen bond interaction between nsp13-2 ZBD and nsp8-2; region 5 includes a faint hydrophobic contact between nsp13-2 1A and nsp12. The movement of nsp13-1 results in distinct variations on its interaction with other components in region 1 and 3, and minor difference in region 2 (Fig. 3c–h and Supplementary Fig. 8); whereas, the interactions in region 4 and region 5 are identical in two forms of mini RTC. In form 1 mini RTC, nsp13-1 employ $_{nsp13-1}$Q194 and $_{nsp13-1}$H320 in the 1B domain to contact with the interface domain of nsp12 in region 1. In region 2, two hydrogen bonds are formed between $_{nsp13-1}$N46 (in ZBD) and $_{nsp8-1}$Q73, $_{nsp13-1}$Y93 (in ZBD) and $_{nsp8-1}$Y71; $_{nsp13-1}$F80 and $_{nsp8-1}$L91 (in ZBD) make hydrophobic contact with $_{nsp8-1}$M70. In region 3, $_{nsp13-2}$T216 binds with the carbonyl oxygen atom of $_{nsp13-1}$R248 and backbone of $_{nsp13-1}$V247, and $_{nsp13-2}$Y217 contact $_{nsp13-1}$E244 with a hydrogen bond. In contrast in form 2 mini RTC, $_{nsp13-1}$H320 and $_{nsp13-1}$Q194 move away from the interface domain of nsp12, losing the two hydrogen bonds in region 1. And in region 3, $_{nsp13-1}$E244 and $_{nsp13-1}$R248 in nsp13-1 1B domain also lost their interactions with $_{nsp13-2}$Y217 and $_{nsp13-2}$T216 of nsp13-2 1B domain.

**Nsp13-1 is essential for the enhanced helicase activity of mini RTC.** Because nsp13-1 1B domain has clear interaction with nsp13-2 1B domain that plays an essential role in RNA template binding, we hypothesize nsp13-1 is indispensable for the function of mini RTC, though no extra density corresponding to the 5′ extension of the template can be found around nsp13-1. We selected two key interacting residues, $_{nsp12}$R365, which mediates nsp12:nsp13-1 interaction, and $_{nsp13-2}$T216, which is involved in nsp13-1:nsp13-2 interaction, to be mutated by alanine residues, and dissected their impacts on the helicase activity (Fig. 1c). Because $_{nsp12}$R365 mediates the interaction of 1B domain of nsp13-1 and interface domain of nsp12 and is not included in the interaction between nsp13-2 and nsp7–nsp8–nsp12, this mutation would abolish the interaction between nsp13-1, but not nsp13-2, with other components in mini RTC. The results show that mini RTC with an $_{nsp12}$R365A mutation has a compared helicase activity with the individual apo nsp13, which is significantly lower than mini RTC. The decrease of helicase activity is observed in mini RTC with an $_{nsp13-1}$T216A mutation. These results suggest that the existence of nsp13-1, as well as the inter-nsp13 interaction at 1B domains, is essential for the enhanced helicase activity of mini RTC.

**Discussion**
Collectively, we propose a model for the helicase-polymerase coupling in the formation of SARS-CoV-2 RTC (Fig. 4). During the formation of mini RTC, nsp13-1 binds with nsp8-1 and nsp12 to provide a basis for the couple of nsp13-2 into mini RTC. The individual nsp13-2 has a basal helicase activity to unwind RNA template with double-strand. After nsp13-2 is assembled into mini RTC, the inter-helicase contact at 1B domains would assist potential structural shift of nsp13-2 1B domain to allow the entrance of its RNA-binding channel in fully open form, as a consequence of the orientation shift of nsp13-1. The unwinded single-strand RNA template passes through the RNA-binding channel of nsp13-2 and extends to the active center of nsp12 for the subsequent nascent nucleic acid synthesis. For note, coronavirus nsp13 has been proposed to unwind RNA in the 5′ to 3′ direction[8,11,17]; however, the 5′ extension of template RNA is fed into the active site of SARS-CoV-2 nsp13 in the 3′ to 5′ direction. RNA polymerase has been known to possess a "backtracked"

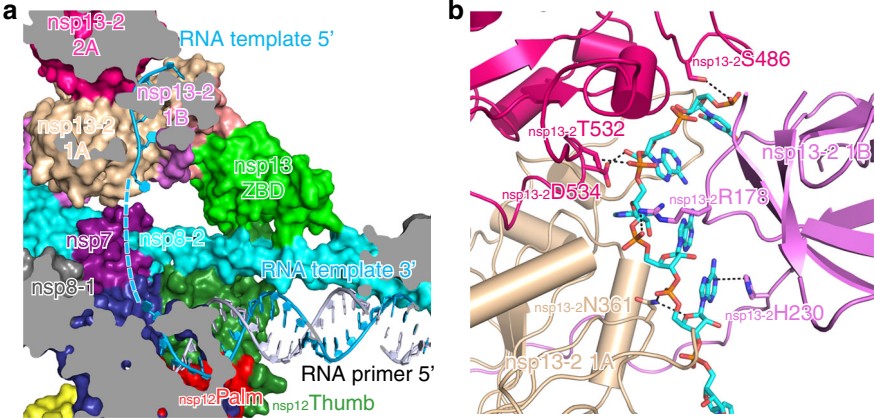

**Fig. 2 Binding of 5′ extension of RNA template in nsp13-2. a** Cutaway view of RNA primer–template pair bound to mini RTC. The structures of nsp7, nsp8, nsp12, and nsp13 are covered by a molecular surface with the same color scheme in Fig. 1. The bound RNA primer–template pair is exhibited as colored cartoons. The missing nucleotides are denoted as a dashed line. **b** Key interactions nsp13-2 and 5′ extension of RNA template. The molecule of nsp13-2 is shown as a cartoon diagram, and key residues involved in the binding with RNA are shown as colored sticks. The dashed lines indicated the bonds with a distance of less than 3.5 Å.

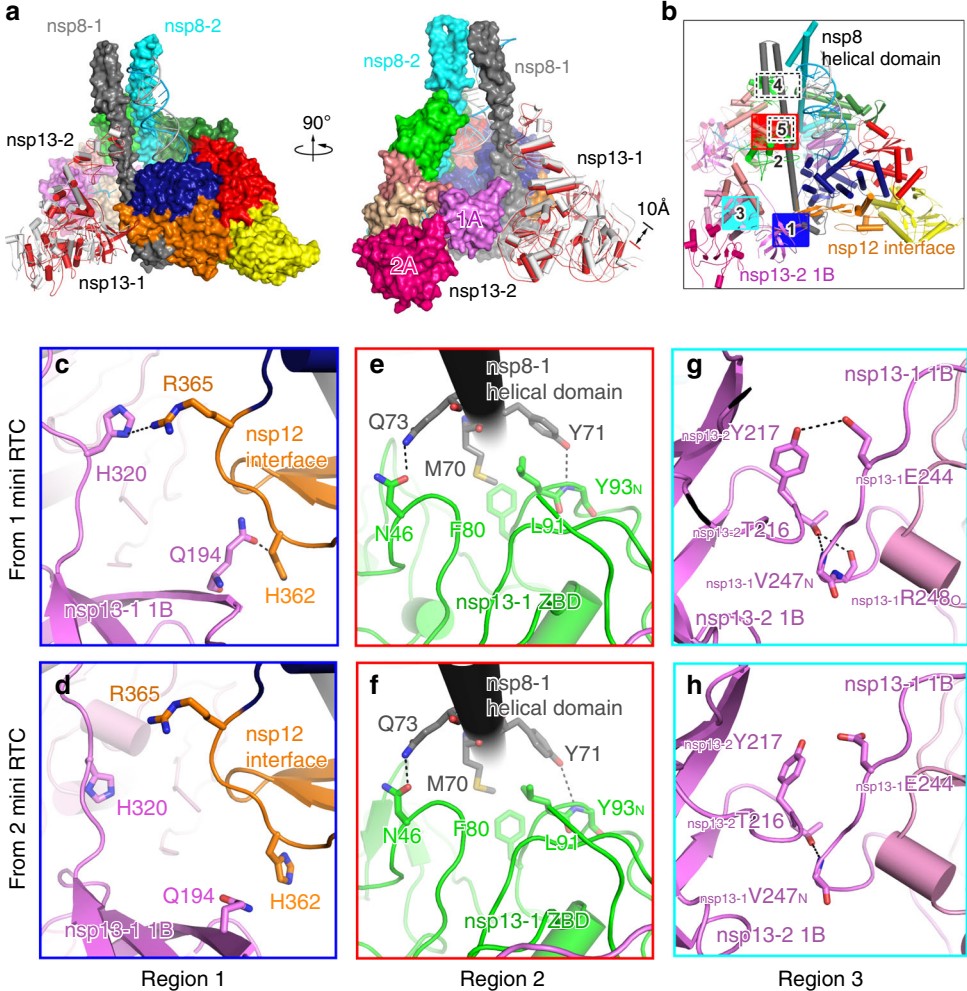

**Fig. 3 Two orientations of nsp13-1. a** Structural comparison of two mini RTCs in two perpendicular views. The structural parts with high similarity in two mini RTCs, including nsp7–nsp8–nsp12-RNA and nsp13-2, are covered by a molecular surface with the same color scheme in Fig. 1. Nsp13-1 molecules in form 1 and form 2 mini RTC are displayed as white and red cartoons. **b** The interaction regions of nsp13 molecules with other components of mini RTC or with each other. The structural details for region 1 (**c**, **d**), region 2 (**e**, **f**) and region 3 (**g**, **h**) are enlarged. Form 1 is in **c**, **e**, **g**; form 2 is in **d**, **f**, **h**. Dashed lines denoted the interactions with a bond distance of less than 3.5 Å.

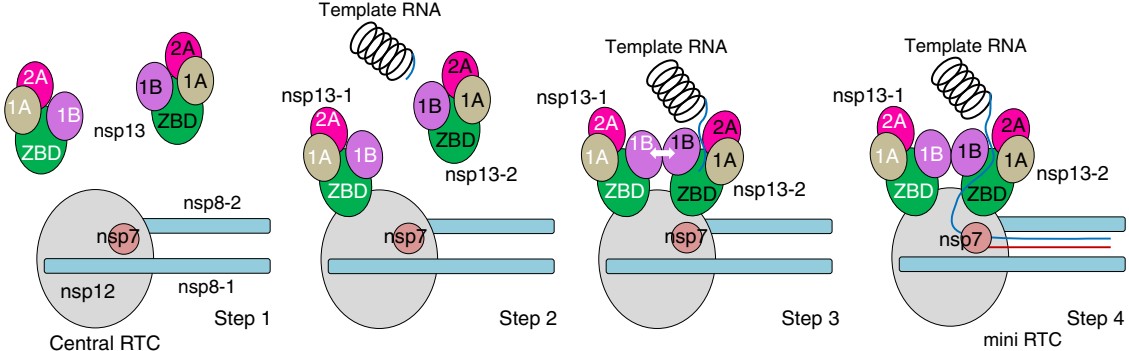

**Fig. 4 A proposed model for the helicase-polymerase coupling in the formation of SARS-CoV-2 RTC.** Nsps composed mini RTC and template-primer RNAs are shown as schematic diagrams. Step 1, nsp12 bound nsp7, and nsp8 to constitute central RTC, waiting for a template RNA unwinding. The nsp13 molecule keeps in an inactive state. Step 2, an nsp13 (nsp13-1) contacts central RTC to form a platform to recruit the nsp13-2. Step 3, nsp13-2 is assembled into mini RTC, and contacts with nsp13-1, allowing the bound RNA translocating towards nsp12 activity center. Step 4, the unwinded single-strand RNA template passes through the RNA-binding channel of nsp13-2 and extends to nsp12 active center for the subsequent nascent RNA synthesis.

feature, in which the productive elongation and translocation complexes are in the same conformation to facilitate reversible backward motion during RNA synthesis[18,19]. It is likely that SARS-CoV-2 mini RTC has the same feature, in which RNA template and primer RNA or product RNA may shift toward the upstream during the nascent RNA synthesis. The structural insight into a mini RTC assembled by SARS-CoV-2 nsp7–nsp8–nsp12–nsp13 provides a basis for further understanding the replication and transcription mechanism of SARS-CoV-2.

While this paper was being prepared, Campbell and colleagues reported their structure of nsp7–8–12–13 in complex with RNA[20]. The structure reported by Campbell's group is overall similar to the structure of the nsp7–8–12–13–RNA complex reported here. In that work, it was proposed that nsp13-2 can be stably bound to polymerase even if nsp13-1 is dissociable and nsp13-1 is likely not necessary for the full function of helicase-polymerase complex[20]. But in this study, we show that nsp13-1 is indispensable for the full function of mini RTC. This difference warrants investigation in a further assembled RTC with more nsps.

## Methods

**Protein production and purification.** The SARS-CoV-2 nsp12 (GenBank: MN908947) gene was cloned into a modified pET-22b vector, with the C-terminus possessing a 10× His-tag (primer information in Supplementary Table 5). The plasmids were transformed into *E. coli* BL21 (DE3), and the transformed cells were cultured at 37 °C in LB media containing 100 mg/L ampicillin. After the OD600 reached 0.8, the culture was cooled to 16 °C and supplemented with 0.5 mM IPTG. After overnight induction, the cells were harvested through centrifugation, and the pellets were resuspended in buffer 1 (20 mM Tris-HCl, pH 8.0, 150 mM NaCl, 4 mM MgCl₂, 10% glycerol) and homogenized with an ultra-high-pressure cell disrupter at 4 °C. The insoluble material was removed through centrifugation at 20,000 × g for 50 min. The fusion protein was incubated with Ni-NTA resin (GE Healthcare, USA) at 4 °C for 1 hour. After eluted by buffer 1 supplemented with 300 mM imidazole, the target protein was further purified by passage through a Hitrap Q ion-exchange column (GE Healthcare, USA) with buffer A (20 mM Tris-HCl, pH 8.0, 4 mM MgCl₂, 10% glycerol, 4 mM DTT) and buffer B (20 mM Tris-HCl, pH 8.0, 1 M NaCl, 4 mM MgCl₂, 10% glycerol, 4 mM DTT). Then it was loaded onto a Superdex 200 10/300 Increase column (GE Healthcare, USA) with buffer 2 (20 mM Tris-HCl, pH 7.5, 250 mM NaCl, and 4 mM MgCl₂). Purified nsp12 was concentrated to 4.8 mg/mL and stored at 4 °C.

The SARS-CoV-2 nsp13 (GenBank: MN908947) gene was inserted into the modified pET-28a vector with a 6× His-tag attached at its N-terminus (primer information in Supplementary Table 5). *E. coli* BL21 (DE3) cells were then transformed by the introduction of this plasmid. Bacteria was induced when OD600 was 0.6–0.8 with 0.2 mM IPTG after cultured in LB medium at 37 °C. After grown at 16 °C for 16–18 h, cells were harvested and resuspended in buffer 3 (20 mM HEPES, pH 7.0, 150 mM NaCl, 4 mM MgCl₂, 10% glycerol). The cell pellets were centrifuged at 20,000 × g for 40 min after lysed by high-pressure homogenization and sonication. After purification by Ni-NTA (Novagen, USA) affinity chromatography, the protein eluted with 200 mM imidazole. The eluate was then further purified by Hitrap S ion-exchange column (GE Healthcare, USA) in a buffer containing 20 mM HEPES, pH 7.0, 50–300 mM NaCl gradient, 2 mM DTT, before loading onto a Superdex 200 10/300 Increase column (GE Healthcare, USA) in the buffer 4 (20 mM HEPES, pH 7.0, 150 mM NaCl, 4 mM MgCl₂, 4 mM DTT). Purified nsp13 was concentrated to 4 mg/mL and stored at 4 °C.

Full-length SARS-CoV-2 nsp7 and nsp8 were co-expressed in *E. coli* BL21 (DE3) cells as a no-tagged protein and a 6 × His-SUMO fusion protein, respectively (primer information in Supplementary Table 5). After purification by Ni-NTA (Novagen, USA) affinity chromatography, the nsp7–nsp8 complex was eluted through on-column tag cleavage by ULP protease. The eluate was further purified by Hitrap Q ion-exchange column (GE Healthcare, USA) and a Superdex 200 10/300 Increase column (GE Healthcare, USA) in a buffer 5 (20 mM Tris-HCl, pH 7.5, 250 mM NaCl, 4 mM MgCl₂).

**Assembly of mini RTC.** First, for assembling stable nsp12–nsp7–nsp8 complex, purified nsp12 with 4.8 mg/mL was incubated with nsp7 and nsp8 at 4 °C for 3h , at a molar ratio of 1:2:2 in a buffer containing 20 mM Tris-HCl, pH 7.5, 250 mM NaCl, and 4 mM MgCl₂. Then the mixture was purification by mono Q 5/50 ion-exchange chromatography (GE Healthcare, USA), and got a stable nsp12–nsp7–nsp8 complex. Annealed RNA scaffold (Fig. 1a) was added to the dialyzed nsp12–nsp7–nsp8 complex at a molar ratio of 1:1 and incubated for 30 min at 25 °C. Then the mixture was incubated with the pretreated nsp13 which was incubated with GDP•BeF₄⁻ for 30 min at 25 °C to assemble mini RTC.

**Nucleic acid unwinding assay.** The nucleic acid unwinding assays were performed as previously reported[9]. Briefly, dsDNA (5′-AATGTCTGACGTAAAGCCTCTAA AATGTCTG-3′-BHQ, CY3-5′-CAGACATTTTAGAGG-3′) was used where the excitation wavelength was set to 547 nm and emission wavelength was set to 562 nm to detect fluorescence of CY3. 200 nM Nsp13 (final concentration) was added to the reaction buffer (50 mM HEPES, pH 7.0, 150 mM NaCl, 4 mM MgCl₂, 0.5 mM EDTA, and 0.1 mg/ml BSA) to incubate with dsDNA and 20 μM trap ssDNA for 5 min. Subsequently, 2 mM ATP (final concentration) was added to initiate the helicase activity, and the fluorescence value was recorded by Perkin-Elmer Enspire plate reader (Perkin-Elmer, USA).

**Cryo-EM sample preparation and data collection.** In total, 3 μL of protein solution at 3 mg/mL (added with 0.025% DDM) was applied onto an H₂/O₂ glow-discharged, 200-mesh Quantifoil R0.6/1.0 grid (Quantifoil, Micro Tools GmbH, Germany). The grid was then blotted for 3.0 s with a blot force of 0 at 8 °C and 100% humidity and plunge-frozen in liquid ethane using a Vitrobot (Thermo Fisher Scientific, USA). Cryo-EM data were collected with a 300 keV Titan Krios electron microscope (Thermo Fisher Scientific, USA) and a K3 direct electron detector (Gatan, USA). Images were recorded at ×29,000 magnification and calibrated super-resolution pixel size 0.82 Å per pixel. The exposure time was set to 2 s with a total accumulated dose of 60 electrons per Å². All images were automatically recorded using SerialEM. A total of 4107 images were collected with a defocus range from −2.0 μm to −1.0 μm. Statistics for data collection and refinement are in Supplementary Table 1.

**Cryo-EM image processing.** All dose-fractioned images were motion-corrected and dose-weighted by MotionCorr2[21] software, and their contrast transfer functions were estimated by ctffind4[22]. A total of 384,727 particles were auto-picked using blob picker and extracted with a box size of 440 pixels in cryoSPARC[23]. The following 2D, 3D classifications, and refinements were all performed in

cryoSPARC. 166,942 particles were selected after two rounds of 2D classification based on the complex integrality. This particle set was used to do Ab-Initio reconstruction in six classes, which were then used as 3D volume templates for heterogeneous refinement, with 109,912 particles converged into one nsp12-nsp7–nsp8–nsp13–RNA complex class. Next, these particles were imported into RELION 3.03[24] to perform local classification to obtain two different conformations. These two conformations are used as an initial model for heterogeneous refinement in cryoSPARC and particles are classified into two particle sets. Finally, non-uniform refinement is applied to the particle sets and result in two maps in final resolution 2.98 Å and 3.84 Å, respectively.

**Model building and refinement**. To solve the structure of the SARS-CoV-2 mini RTC complex, the structure of the SARS-CoV-2 nsp12 and nsp7-8 complex (PDB: 7BTF) and SARS-CoV-2 nsp13 (PDB: 6ZSL) were individually placed and rigid-body fitted into the cryo-EM map using UCSF Chimera[25]. The model was manually built in Coot[26] with the guidance of the cryo-EM map and in combination with real space refinement using Phenix[27]. The data validation statistics are shown in Supplementary Table 1.

**Reporting summary**. Further information on research design is available in the Nature Research Reporting Summary linked to this article.

## Data availability
The cryo-EM density maps and the structures were deposited into the Electron Microscopy Data Bank (EMDB) and Protein Data Bank (PDB) with the accession numbers: form 1, EMD-30492 and 7CXM; form 2, EMD-30493 and 7CXN. Other data are available from the corresponding authors upon reasonable request. Source data are provided with this paper.

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

## Acknowledgements
We would like to thank the Bio-Electron Microscopy Facility of ShanghaiTech University for the data collection. This work was supported by the National Program on Key Research Project of China (2020YFA0707500 and 2017YFC0840300) and Tsinghua University Spring Breeze Fund.

## Author contributions
Z.R. and Z.L. conceived the project and designed the experiments. L.Y., Y.Z., J.G., L.Z., Z.J., Y.G., T.W., M.L., and H.W. performed experiments. L.Y., Y.H., Q.W., Z.L., and Z.R. analyzed the data. L.Y., Z.L., and Z.R. wrote the paper. All authors discussed the experiments, read, and approved the paper.

## Competing interests
The authors declare no competing interests.
