## [Peer Review File · Nature Communications]

REVIEWER COMMENTS

Reviewer #1 (Remarks to the Author):

In this manuscript, Yan et al. describe the structure of a SARS-CoV-2 RNA-dependent RNA polymerase complex bound to a duplex of template- and product RNA and the helicase nsp13, a complex they call "mini RTC". The structure shows how two copies of nsp13 interact with the RdRp complex and one of these two copies also with the template RNA. In the latter, the single-stranded template RNA bound to nsp13 is visible. Within the structural data, the authors identify two particle populations that correspond to compositionally identical complexes, but differ in the orientation of the nsp13 monomers relative to each other. The structural data appears to be of good quality, although the refinement statistics indicate that the models could be likely be improved with some effort.

Overall, the authors present an important structure that shows how the helicase nsp13 interacts with the RdRp complex. This is an important step forward in our understanding of coronavirus biology, because the role of the nsp13 in viral replication is not well understood. That said, the novelty of this work is somewhat limited by the fact that an essentially identical structure was recently published Chen et al., Cell 2020, which the authors do not cite. I believe this manuscript would profit by including a reference to this paper and also pointing out differences between the studies. I have a few concerns that I believe should be addressed before publication can be considered.

Major concerns

- As mentioned above, an essentially identical structure of the RdRp-nsp13 complex was recently published (Chen et al., Cell 2020). This manuscript was posted on bioRxiv on July 13th, and was published as in press corrected proof by Cell on July 28th. Unfortunately, the authors miss the opportunity to mention and cite this work and to discuss their results in the context of this previous paper, which I think would be appropriate. There are a number of interesting differences between the two studies. For example, Chen et al. observed subpopulations of different stoichiometries (one or two nsp13 copies bound). Did the authors of this work also observe a population with only one nsp13 copy bound? This would be relevant also to their proposed mechanism of sequential binding of the two nsp13s. (On a side note: the nomenclature for the two nsp13 in this manuscript is identical to the one in Chen et al., but inverse. So nsp13-1 in this manuscript corresponds to nsp13-2 in Chen et al., and vice versa).

- The authors describe and also model how the downstream template RNA binds to nsp13. This was not seen in sufficient detail to model bases in the previous study by Chen et al., and thus provides novelty. The authors should discuss the potential functional implications of this binding: There are other structures of SF1B helicases with substrate RNA bound. Is the binding mode observed in nsp13 "canonical" or does it differ? Are the residues that are proposed to interact with RNA conserved across nsp13 sequences in different nidoviruses? Most importantly, does the RNA binding allow to deduce a hypothesis on the function of nsp13? The authors speculate that the unwound strand passes through the binding channel of nsp13 to the active center of nsp12 (line 163-165). However, nsp13 has been proposed to unwind RNA in the 5' to 3' direction, yet the template strand is fed into the active site in the 3' to 5' direction. Does the structure provide any clue to this paradox? It is noteworthy that the helicase assay presented in Fig 1 also monitors 5' to 3' duplex unwinding. While it would be beyond the scope of this work to probe whether nsp13 can actually unwind in the opposite direction when in the mini RTC, the authors should discuss this in the context of their model.

- Table S1: I believe that the models could be somewhat improved. A clashscore of >20 and >10% allowed Ramachandran residues are rather poor, at least for the higher resolution model. Judging from the validation report and local resolution map, I suspect many of the poor geometry regions in one of the nsp13 copies. Since a good starting model (7ZSF) is available, I think this could be easily improved with appropriate refinement restraints.

Minor points

- Line 41-44: This sentence is confusing, because WHO is mentioned twice. I think it would be easier if it read "According to the WHO report..."
- Line 61: "reconstituted" instead of "constituted"
- Line 118: nsp13-1 twice – one should be nsp13-2?
- The authors state that nsp13 was pre-incubated with the transition state analog GDP-BeF₄. Was there a particular reason to choose GDP over ADP? Do the authors see a density for this nucleotide bound to nsp13? What state of the catalytic cycle of helicases do the observed structures resemble?
- Supplementary Figure 2: The angular distribution, B factor and FSC plots should be given for both structures discussed (Conformation I and II)
- Supplementary Figure 2: The processing scheme lacks a few steps that are described in the methods, namely the classification in Relion and subsequent import and refinement in cryoSPARC.
- Supplementary Figure 4: The model shown in a) and b) for the template RNA bound to nsp13 does not correspond to the sequence given: UAAAU. The model appears to be all U. Did the authors model this section as poly-U instead of the actual sequence? If yes, they should state this in the main text and in Figure legend 2b. The numbering of the sequence also does not appear right. Nucleotide +8 to +12 should be AAAAU, according to Figure 1a. Also, the labels in Supplementary Figure 4a and b for this region are "p", while I believe they should be "t" (for template) and the color should be grey not blue.
- Supplementary Figure 6b: What parts of the protein were the structures aligned with?
- Table S1: Map resolution range has negative values. Please correct.
- Supplementary Table 4: The legend or table should state that the HDX data is cited from another study.

Reviewer #2 (Remarks to the Author):

Yan et al. reported cryo-EM structures of two forms of SARS-CoV-2 mini replication and transcription complex (RTC), which consists of RNA dependent RNA polymerase (RdRp, nsp12), cofactors nsp7 and nsp8, and two helicases (nsp13-1 and nsp13-2) with a template-primer RNA. Yan et al. also analyzed the interface between nsp13 and nsp12 by mutagenesis and helicase activity. Previous apo nsp7-nsp8-nsp12 (Gao et al., 2020) and nsp7-nsp8-nsp12 (Wang et al., 2020) with RNA has been determined by the co-author group.

Yan et al. used a combination of biochemical and biophysical methods that include cryo-EM, mutagenesis, HDX-MS, activity assay, etc. Yan et al. showed that the different conformations for two copies of nsp13. Yan et al. proposed a mechanistic model that is consistent with the data.

Overall, it is a paper with solid experimental data. The reported data revealed the conformational shift of nsp13 between two forms of mini-RTC, highlighted the role of nsp13 involved in the helicase activity of mini-RTC. However, there are still some issues regarding the conclusion which need to be addressed:

1. The title says replication and transcription complex. Is it the same complex work for the replication and transcription? Do they have any differences?
2. There has another paper talking about the complex of Nsp7/8/12 and Nsp13 (Chen et al., Structural Basis for Helicase-Polymerase Coupling in the SARS-CoV-2 Replication-Transcription Complex, Cell (2020), PDB ID: 6XEZ), which should be cited and compared with. What is the difference and what is new?
3. In the method of "assembly of mini RTC", a lot details are missing, such as the concentration of Nsp12 and annealed RNA during assembling, the buffer for mono-Q, and was the sample diluted before loaded to mono-Q, "dialyzed nsp12-nsp7-nsp8" mean dialyzed to what buffer?

4. For the nsp13 preparation. How Nsp13 pretreated with GDP-BeF4-? how much Nsp13 added to the Nsp1/8/12 complex (concentration? Ratio?)
5. Figure S1: the gel band for nsp7 is not clear.
6. It's not clear about the reason of switching from cryoSPARC to RELION, and then back to cryoSPARC during the 3D reconstruction. Are there any particular reasons?
7. Figure S2: panel e, FSC curves. The masked and loose or tight mask has a big jump on the resolution. Is there any sign of over-refinement?
8. Figure S4: what are the sigma levels used for displaying the map?
9. The helicase activity test results for R365 and T216 mutants are not shown.
10. The HDX-MS raw data is missing.
11. It will be better if a cartoon or illustration of the proposed model is shown?
12. What are the roles of two different conformations of nsp13-1?
13. Lane 270, the PDB, AND EMDB NUMBER are missing.

Point-to-point responses to comments

Response to Reviewer #1:

In this manuscript, Yan et al. describe the structure of a SARS-CoV-2 RNA-dependent RNA polymerase complex bound to a duplex of template- and product RNA and the helicase nsp13, a complex they call “mini RTC”. The structure shows how two copies of nsp13 interact with the RdRp complex and one of these two copies also with the template RNA. In the latter, the single-stranded template RNA bound to nsp13 is visible. Within the structural data, the authors identify two particle populations that correspond to compositionally identical complexes, but differ in the orientation of the nsp13 monomers relative to each other. The structural data appears to be of good quality, although the refinement statistics indicate that the models could be likely be improved with some effort.

Overall, the authors present an important structure that shows how the helicase nsp13 interacts with the RdRp complex. This is an important step forward in our understanding of coronavirus biology, because the role of the nsp13 in viral replication is not well understood. That said, the novelty of this work is somewhat limited by the fact that an essentially identical structure was recently published Chen et al., Cell 2020, which the authors do not cite. I believe this manuscript would profit by including a reference to this paper and also pointing out differences between the studies. I have a few concerns that I believe should be addressed before publication can be considered.

Major concerns

- As mentioned above, an essentially identical structure of the RdRp-nsp13 complex was recently published (Chen et al., Cell 2020). This manuscript was posted on bioRxiv on July 13th, and was published as in press corrected proof by Cell on July 28th. Unfortunately, the authors miss the opportunity to mention and cite this work and

to discuss their results in the context of this previous paper, which I think would be appropriate.

Response: Sorry for missing this citation. We provide the correct reference and state this in Discussion section as following. “While this manuscript was being prepared, Campbell and colleagues reported their structure of nsp7-8-12-13 in complex with RNA. The structure reported by Campbell’s group is overall similar to the structure of nsp7-8-12-13-RNA complex reported here. In that work, it was proposed that nsp13-2 can be stably bound to polymerase even if nsp13-1 is dissociable and nsp13-1 is likely not necessary for the full function of helicase-polymerase complex. But in this study, we show that nsp13-1 is indispensable for the full function of mini RTC. This difference warrants investigation in a further assembled RTC with more nsps. “

There are a number of interesting differences between the two studies. For example, Chen et al. observed subpopulations of different stoichiometries (one or two nsp13 copies bound). Did the authors of this work also observe a population with only one nsp13 copy bound? This would be relevant also to their proposed mechanism of sequential binding of the two nsp13s. (On a side note: the nomenclature for the two nsp13 in this manuscript is identical to the one in Chen et al., but inverse. So nsp13-1 in this manuscript corresponds to nsp13-2 in Chen et al., and vice versa).

Response: We thank the reviewer for raising this crucial question. After removing impurities by 2D classification, several iterations of ab-initio reconstruction and heterogeneous refinement are applied to particles to classify the different conformations. Classification showed that the classes with two nsp13 bounded were dominant (94%) and the other classes (6%) were impurities, and the conformation which has one nsp13 is not observed in all classification steps. Compared the method of our mini RTC assembly with Dr. Campbell’s method, we found they add nsp13 into RTC at a molar ratio of 1:1, but ours method is 2:1. We think the different molar ratio of nsp7-8-12 and nsp13 causes this difference.

We named two nsp13 molecules according to their interacting nsp8 molecules, which have been named in our previously reported nsp7-8-12 complex (Gao, Science, 2020). Nsp13 interacting with nsp8-1 is named as nsp13-1, and another one is named as nsp8-2. This causes the inversed nomenclature compared with Dr. Compbell's work.

- The authors describe and also model how the downstream template RNA binds to nsp13. This was not seen in sufficient detail to model bases in the previous study by Chen et al., and thus provides novelty.

Response: We will add this model into Figure 4 in revision manuscript.

The authors should discuss the potential functional implications of this binding: There are other structures of SF1B helicases with substrate RNA bound. Is the binding mode observed in nsp13 “canonical” or does it differ? Are the residues that are proposed to interact with RNA conserved across nsp13 sequences in different nidoviruses?

Response: We will expand the discussion according to this comment.

“RecD2 is a member of SF1B with low sequence homology with SARS-CoV-2 nsp13. But the RNA binding mode of SARS-CoV-2 nsp13 is conserved with that in SF1B family. Sequence comparison of the key residues for RNA binding in SARS-CoV-2/SARS-CoV/MERS-CoV nsp13 and EAV nsp9 (helicase) show that R178 (in 1B domain) and T532/D534 (in 2A domain) are highly conserved. Moreover,

H230 (in 1B domain) and N361 (in 1A domain) are conserved in coronaviruses but are different with their counterparts in EAV (Supplementary Figure 7).” A supplementary figure 7 is also provided in the supplementary information.

Most importantly, does the RNA binding allow to deduce a hypothesis on the function of nsp13? The authors speculate that the unwound strand passes through the binding channel of nsp13 to the active center of nsp12 (line 163-165). However, nsp13 has been proposed to unwind RNA in the 5' to 3' direction, yet the template strand is fed into the active site in the 3' to 5' direction. Does the structure provide any clue to this

paradox? It is noteworthy that the helicase assay presented in Fig 1 also monitors 5' to 3' duplex unwinding. While it would be beyond the scope of this work to probe whether nsp13 can actually unwind in the opposite direction when in the mini RTC, the authors should discuss this in the context of their model.

Response: We thank this suggestion and discuss this in Discussion section.

“For note, coronavirus nsp13 has been proposed to unwind RNA in the 5' to 3' direction; however, the 5' extension of template RNA is fed into the active site of SARS-CoV-2 nsp13 in the 3' to 5' direction. RNA polymerase has been known to possess a “backtracked” feature, in which the productive elongation and translocation complexes are in the same conformation to facilitate reversible backward motion during RNA synthesis. It is likely that SARS-CoV-2 mini RTC has the same feature, in which RNA template and primer RNA or product RNA may shift towards the upstream during the nascent RNA synthesis.”

- Table S1: I believe that the models could be somewhat improved. A clashscore of >20 and >10% allowed Ramachandran residues are rather poor, at least for the higher resolution model. Judging from the validation report and local resolution map, I suspect many of the poor geometry regions in one of the nsp13 copies. Since a good starting model (7ZSF) is available, I think this could be easily improved with appropriate refinement restraints.

Response: We appreciate the review's suggestion and rebuild the model using 7ZSF as a starting model, and the clashscore and Ramachandran have been optimized.

Minor points

- Line 41-44: This sentence is confusing, because WHO is mentioned twice. I think it would be easier if it read “According to the WHO report....”

Response: This sentence is modified as “According to the World Health Organization (WHO) on August 6th, 2020, 2.04 million infections and over 744 thousand deaths have been confirmed globally”.

- Line 61: “reconstituted” instead of “constituted”

Response: Modified.

- Line 118: nsp13-1 twice – one should be nsp13-2?

Response: This mistake is corrected. Thank you!

- The authors state that nsp13 was pre-incubated with the transition state analog GDP-BeF₄. Was there a particular reason to choose GDP over ADP? Do the authors see a density for this nucleotide bound to nsp13? What state of the catalytic cycle of helicases do the observed structures resemble?

Response: In our long-term work in the study of coronavirus nsps, we found that the recombinant polymerase nsp12 and helicase nsp13 proteins are unstable in the purified condition in our lab. We tested the stability of nsp12 and nsp13 by thermal stability assay, and found GDP can increase their stability. Therefore, we use GDP.BeF⁻ buffer to treat both proteins before further assembly.

- Supplementary Figure 2: The angular distribution, B factor and FSC plots should be given for both structures discussed (Conformation I and II)

Response: We provide these information in Supplementary Figure 2. The golden-standard FSC plots (Supplementary Figure 2e), the angular distributions (Supplementary Figure 2f), the Guinier plots for B-factor estimation (Supplementary Figure 2g) and the directional FSC plots (Supplementary Figure 2h) for both conformations are added in Supplementary Figure 2.

- Supplementary Figure 2: The processing scheme lacks a few steps that are described in the methods, namely the classification in Relion and subsequent import and refinement in cryoSPARC.

Response: After Hetero refinement in cryoSPARC, we were able to obtain a particle set corresponding to nsp12-nsp7-nsp8-nsp13-RNA class. Judging by the local densities, there might be some flexible differences between the two nsp13 molecules and we would like to perform local classifications on this region. CryoSPARC could not perform local classifications with a mask while RELION could perform this. Thus, we imported the particle set into RELION 3.03 and finally obtained two 3D volumes with different conformations. In the algorithm of cryoSPARC, it deeply relied on the initial 3D volume references and once we provided the correct 3D references, it could help us push the final resolution. The same method was used in the article "Structural Basis for RNA Replication by the SARS-CoV-2 Polymerase" (Wang, Cell, 2020)

- Supplementary Figure 4: The model shown in a) and b) for the template RNA bound to nsp13 does not correspond to the sequence given: UAAAU. The model appears to be all U. Did the authors model this section as poly-U instead of the actual sequence? If yes, they should state this in the main text and in Figure legend 2b. The numbering of the sequence also does not appear right. Nucleotide +8 to +12 should be AAAAU, according to Figure 1a. Also, the labels in Supplementary Figure 4a and b for this

region are “p”, while I believe they should be “t” (for template) and the color should be grey not blue.

Response: Thank the reviewer for raising this question. When build the complex structure, we found the density map of template RNA is discontinuous from nsp12 activity center to nsp13-2, which make hard to build precise nucleotide number, and according to structural biology convention, we model the RNA map bound to nsp13-2 as poly-U. As suggested, nucleotide +8 to +12 RNA sequence was fitted into density map in Figure 2a. We apologize the reviewer for this mistake in Supplementary Figure 4a and b, and have corrected them as suggested.

- Supplementary Figure 6b: What parts of the protein were the structures aligned with?

Response: Nsp13 apo structure determined by X-ray crystal, nsp13-1 and nsp13-2 are two molecules from min RTC form 1 were used to generate this figure. We superposed these structures and found the 1B domains in three molecules have conformational changes.

- Table S1: Map resolution range has negative values. Please correct.

Response: We have corrected it as suggested. Thank you.

- Supplementary Table 4: The legend or table should state that the HDX data is cited from another study.

Response: We thank the reviewer for this suggestion and have added the statement in Supplementary Table 4.

Response to Reviewer #2:

Yan et al. reported cryo-EM structures of two forms of SARS-CoV-2 mini replication and transcription complex (RTC), which consists of RNA dependent RNA polymerase (RdRp, nsp12), cofactors nsp7 and nsp8, and two helicases (nsp13-1 and nsp13-2) with a template-primer RNA. Yan et al. also analyzed the interface between nsp13 and nsp12 by mutagenesis and helicase activity. Previous apo nsp7-nsp8-nsp12(Gao et al., 2020) and nsp7-nsp8-nsp12(Wang et al., 2020) with RNA has been determined by the co-author group.

Yan et al. used a combination of biochemical and biophysical methods that include cryo-EM, mutagenesis, HDX-MS, activity assay, etc. Yan et al. showed that the different conformations for two copies of nsp13. Yan et al. proposed a mechanistic model that is consistent with the data.

Overall, it is a paper with solid experimental data. The reported data revealed the conformational shift of nsp13 between two forms of mini-RTC, highlighted the role of nsp13 involved in the helicase activity of mini-RTC. However, there are still some issues regarding the conclusion which need to be addressed:

1. The title says replication and transcription complex. Is it the same complex work for the replication and transcription? Do they have any differences?

Response: For coronavirus, the non-structural proteins (nsps) assemble a so-called “replication and transcription complex” to facilitate the entire transcription and replication steps within virus lifecycle. The exact architecture of RTC in each reaction steps are not well known at this moment. But in our opinion, the overall architecture of RTC should keep consistent within replication and transcription steps.

2. There has another paper talking about the complex of Nsp7/8/12 and Nsp13 (Chen et al., Structural Basis for Helicase-Polymerase Coupling in the SARS-CoV-2 Replication-Transcription Complex, Cell (2020), PDB ID: 6XEZ), which should be cited and compared with. What is the difference and what is new?

Response: The work published by Dr. Compbell's group is cited and compared with our work in this revision. In this study, we determined the structure of nsp12-nsp7-nsp8₂-nsp13₂-RNA, with form 1 and form 2 conformations and found nsp13-1 stabilizes the overall architecture of mini RTC by contacting with nsp13-2, which anchors the 5' extension of RNA template, as well as interacting with nsp7-nsp8-nsp12-RNA complex. We also show that nsp13-1 is essential for the enhanced helicase activity of mini RTC. In contrast, the work recently published by Dr. Compbell's group (Chen, Cell 2020) proposed that only one nsp13 molecule appears to be stably bound to polymerase and is required for the full function of SARS-CoV-2 RTC, whereas nsp13-1 is dissociable.

3. In the method of "assembly of mini RTC", a lot details are missing, such as the concentration of Nsp12 and annealed RNA during assembling, the buffer for mono-Q, and was the sample diluted before loaded to mono-Q, "dialyzed nsp12-nsp7-nsp8" mean dialyzed to what buffer?

Response: We apologize for missing these information and provide more details in the revision.

4. For the nsp13 preparation. How Nsp13 pretreated with GDP-BeF₄? how much Nsp13 added to the Nsp1/8/12 complex (concentration? Ratio?)

Response: From SARS-CoV to SARS-CoV-2, the helicase nsp13, is very unstable and easy to precipitate, we found the GDP.BeF⁻ can enhance the stability of nsp13. When nsp13 sample is purified by Superdex 200, we concentrate nsp13 to 6.7 mg/ml for the volume of 250 µl, and dilute the final volume for 2.5ml using the buffer (50 mM

Hepes pH 7.0, 100 mM NaCl, 4 mM MgCl₂, 2 mM GDP, 2 mM BeSO₄, 20 mM NaF) in 30 °C for 30 min, and concentrate to 250 µl, add to the nsp7/8/12 at a molar ratio of 2:1. (For example, the weight of nsp7/8/12 is about 170 kDa, and the concentration is 9 mg/ml, we mix nsp13 and nsp7/8/12 at a volume ratio of 1:1).

5. Figure S1: the gel band for nsp7 is not clear.

Response: We replace the old Supplementary Figure 1 by a new one in which nsp7 is clear.

6. It's not clear about the reason of switching from cryoSPARC to RELION, and then back to cryoSPARC during the 3D reconstruction. Are there any particular reasons?

Response: After Hetero refinement in cryoSPARC, we were able to obtain a particle set corresponding to nsp12-nsp7-nsp8-nsp13-RNA class. Judging by the local densities, there might be some flexible differences between the two nsp13 molecules and we would like to perform local classifications on this region. CryoSPARC could not perform local classifications with a mask while RELION could perform this. Thus, we imported the particle set into RELION 3.03 and finally obtained two 3D volumes

with different conformations. In the algorithm of cryoSPARC, it deeply relied on the initial 3D volume references and once we provided the correct 3D references, it could help us push the final resolution. The same method was used in the article "Structural Basis for RNA Replication by the SARS-CoV-2 Polymerase" (Wang, Cell, 2020)

7. Figure S2: panel e, FSC curves. The masked and loose or tight mask has a big jump on the resolution. Is there any sign of over-refinement?

Response: The "big jump ", also called "downward" in FSC is common in the calculations from cryoSPARC and it is depended on the algorithm of cryoSPARC. Even for the high quality maps, the same phenomenon may also occur. Judging by the results of local resolution distribution and 3D FSC histogram, we think that there is no obvious overfitting in the refinement process. Moreover, in the FSC curves, the final corrected FSC shows quite well, indicating the final reconstruction is not questionable. As for the jump in the resolution, we suppose it may be relevant to low SNR at some specified space frequencies (As the data was collected at a defocus range).

8. Figure S4: what are the sigma levels used for displaying the map?

Response: The sigma levels are provided in the legends.

9. The helicase activity test results for R365 and T216 mutants are not shown.

Response: The helicase activity of R365A and T216A are shown in Figure 1c (mut1 is R365A, mut2 for T216A). We replace the label for clear understanding. Thank you.

10. The HDX-MS raw data is missing.

Response: We thank the reviewer for pointing this out, HDX-MS data is cited from Jia Z and Yan L (NAR, 2019). We explain this in the footnote of Supplementary Table 4.

11. It will be better if a cartoon or illustration of the proposed model is shown?

Response: We made a model in Figure 4.

12. What are the roles of two different conformations of nsp13-1?

Response: nsp13-1 in form 1 involves in two interfaces which are shown in Figure 3. Region 1 includes the 1B interaction between nsp13-1 and nsp13-2, and region 3 includes the interaction between nsp13-1 1B domain and nsp12 Interface. In form 1, the interactions of region 1 and region 3 are stronger than that in form 2, and the hypothesis is that nsp13-1 is active in form 1 and inactive in form 2, the conformation shifts from form 1 to form 2 decide the helicase activity of nsp13-2.

13. Lane 270, the PDB, AND EMD NUMBER are missing.

Response: The accession numbers are provided in the revision.

REVIEWERS' COMMENTS

Reviewer #1 (Remarks to the Author):

The authors have addressed all my concerns and improved the manuscript. In my opinion, the manuscript can be accepted for publication after addressing these remaining points:

- Line 47: nsp7 and nsp8 are referred to as co-factors; I suggest a different term, for example "auxiliary factors", because cofactors are defined in biochemistry as non-proteinaceous compounds or ions required for catalysis
- Line 100: Please include that the RNA region connecting the RdRp active site and the nsp13-2 active site was not modeled due to insufficient map quality, as shown in Figure 2 (dashed line).
- Line 131: "mini" instead of "min"
- Line 123: The first sentence to this paragraph is a bit confusing: It starts with RecD2, but this protein is neither in the alignment nor discussed? Do the authors mean that RecD2 is a prototype SF1B helicase and similar to nsp13? If so, why is it not in the alignment? Or are all the helicases in the alignment similar to RecD2?
- Figure 4: This is a very neat figure. However, I have two concerns regarding this. Firstly, in the discussion the authors describe a model in which binding of one nsp13 facilitates binding and conformational activation of the second nsp13. This is a very plausible model based on their data. In the figure, however, the authors imply that nsp13-2 binds RNA first, before the nsp13-2-RNA complex binds nsp12-7-8-13. To my knowledge, there is no data that would justify the assumption that RNA first binds to nsp13 before binding to the core RdRp complex. I therefore suggest modifying Figure 4 to include both possibilities: RNA binding to RdRp first, or RNA binding to nsp13 first. The core part of the model should be what is supported by the data in this manuscript, which is that two nsp13 copies bind to core RdRp and nsp13-1 modulates the activity of nsp13-2. This would also resolve my second concern, which is that in the sequence of events depicted in the current Figure 4, nsp13-2 would need to unwind in the 3' to 5' direction to feed the single-stranded template into the active site, which contradicts previous biochemistry. If nsp13-2 binds free RNA before it binds RdRp, it is not obvious why its directionality should be reversed. These changes could be accompanied by a clarification in the discussion that the authors hypothesize that one of the functions of nsp13 may be to facilitate backtracking.
- Table S1: Model resolution range is lacking a character for both models
- As requested, the authors have improved the model clashscores. In my opinion, the Ramachandran allowed vs. favored are still quite high, at least for the 3Å resolution structure. This is, however, acceptable if the authors wish to publish as is.
- The manuscript would benefit by some careful grammar proofreading, which should be done after acceptance.

Point-to-point responses to comments

Response to Reviewer #1:

Reviewer #1 (Remarks to the Author):

The authors have addressed all my concerns and improved the manuscript. In my opinion, the manuscript can be accepted for publication after addressing these remaining points:

- Line 47: nsp7 and nsp8 are referred to as co-factors; I suggest a different term, for example “auxiliary factors”, because cofactors are defined in biochemistry as non-proteinaceous compounds or ions required for catalysis.

Response: We made this modification.

- Line 100: Please include that the RNA region connecting the RdRp active site and the nsp13-2 active site was not modeled due to insufficient map quality, as shown in Figure 2 (dashed line).

Response: We provided the statement accordingly.

- Line 131: “mini” instead of “min”

Response: We have corrected it.

- Line 123: The first sentence to this paragraph is a bit confusing: It starts with RecD2, but this protein is neither in the alignment nor discussed? Do the authors mean that RecD2 is a prototype SF1B helicase and similar to nsp13? If so, why is it not in the alignment? Or are all the helicases in the alignment similar to RecD2?

Response: We thank the reviewer for pointing it out, and have added the RecD2 sequence alignment in Supplementary Figure 7. Though sequence similarity in SARS-CoV-2/SARS-CoV/MERS-CoV nsp13 and EAV helicase (nsp9) and RecD2 are very low, key residues R178 (in 1B domain) and T532 (in 2A) for RNA bound are highly conservative.

- Figure 4: This is a very neat figure. However, I have two concerns regarding this. Firstly, in the discussion the authors describe a model in which binding of one nsp13 facilitates binding and conformational activation of the second nsp13. This is a very plausible model based on their data. In the figure, however, the authors imply that nsp13-2 binds RNA first, before the nsp13-2-RNA complex binds nsp12-7-8-13. To my knowledge, there is no data that would justify the assumption that RNA first binds to nsp13 before binding to the core RdRp complex. I therefore suggest modifying

Figure 4 to include both possibilities: RNA binding to RdRp first, or RNA binding to nsp13 first. The core part of the model should be what is supported by the data in this manuscript, which is that two nsp13 copies bind to core RdRp and nsp13-1 modulates the activity of nsp13-2. This would also resolve my second concern, which is that in the sequence of events depicted in the current Figure 4, nsp13-2 would need to unwind in the 3' to 5' direction to feed the single-stranded template into the active site, which contradicts previous biochemistry. If nsp13-2 binds free RNA before it binds RdRp, it is not obvious why its directionality should be reversed. These changes could be accompanied by a clarification in the discussion that the authors hypothesize that one of the functions of nsp13 may be to facilitate backtracking.

Response: We thank this suggestion. We modified the model as following, in which two nsp13 copies bind to RdRp first and then the template RNA binds to nsp13-2 which unwinds RNA secondary structures to feed the single-stranded template into the active site of RdRp.

- Table S1: Model resolution range is lacking a character for both models.

Response: We corrected them.

- As requested, the authors have improved the model clashscores. In my opinion, the

Ramachandran allowed vs. favored are still quite high, at least for the 3Å resolution structure. This is, however, acceptable if the authors wish to publish as is.

Response: We further improved our model, in which the Ramachandran favored of form 1 and form 2 are for 92.04% and 90.1%, respectively. New validation reports are attached with this submission. Thanks!

- The manuscript would benefit by some careful grammar proofreading, which should be done after acceptance.

Response: We have very carefully checked the grammatical issue to solve some problem.